# Mean-Field Langevin Dynamics : Exponential Convergence and Annealing

**Lénaïc Chizat** *lenaic.chizat@epfl.ch*
*EPFL*

**Reviewed on OpenReview:** *https://openreview.net/forum?id=BDqzLH1gEm*

## Abstract

*Noisy particle gradient descent* (NPGD) is an algorithm to minimize convex functions over the space of measures that include an entropy term. In the many-particle limit, this algorithm is described by a *Mean-Field Langevin* dynamics—a generalization of the Langevin dynamic with a non-linear drift—which is our main object of study. Previous work have shown its convergence to the unique minimizer via non-quantitative arguments. We prove that this dynamics converges at an exponential rate, under the assumption that a certain family of Log-Sobolev inequalities holds. This assumption holds for instance for the minimization of the risk of certain two-layer neural networks, where NPGD is equivalent to standard noisy gradient descent. We also study the annealed dynamics, and show that for a noise decaying at a logarithmic rate, the dynamics converges in value to the global minimizer of the unregularized objective function.

## 1 Introduction

Let $\mathcal{P}_2(\mathbb{R}^d)$ (resp. $\mathcal{P}_2^a(\mathbb{R}^d)$) be the set of probability measures (resp. absolutely continuous probability measures) with finite second moment on $\mathbb{R}^d$ and let $G : \mathcal{P}_2(\mathbb{R}^d) \to \mathbb{R}$ be a convex function which is "smooth" in the sense of Assumption 1 below. Our goal is to solve problems of the form

$$\min_{\mu \in \mathcal{P}_2^a(\mathbb{R}^d)} F_\tau(\mu) \quad \text{where} \quad F_\tau(\mu) := G(\mu) + \tau H(\mu) \tag{1}$$

with $H(\mu) := \int \log(\frac{\mathrm{d}\mu}{\mathrm{d}x})\mathrm{d}\mu$ the entropy of $\mu$ and $\tau > 0$ the regularization/temperature parameter. See Section 5 for examples of problems of this form (typically with $\tau = 0$) that arise in machine learning such as the regularized risk functional of wide two-layer neural networks, or Maximum Mean Discrepancy (MMD) minimization.

**Noisy Particle Gradient Descent (NPGD)** The starting idea of NPGD is to parameterize the measure $\mu$ as a mixture of $m$ particles $\mu = \frac{1}{m} \sum_{i=1}^m \delta_{X_i}$. Let $\mathbf{X} = (X_1, \ldots, X_m) \in (\mathbb{R}^d)^m$ encode the position of all particles and consider the function

$$G_m(\mathbf{X}) := G\Big(\frac{1}{m} \sum_{i=1}^m \delta_{X_i}\Big). \tag{2}$$

Then, NPGD is just noisy gradient descent on $G_m$ with initialization sampled from $\mu_0 \in \mathcal{P}_2(\mathbb{R}^d)$. It is defined, for $k \geq 0$, as

$$\mathbf{X}[k+1] = \mathbf{X}[k] - m\eta \nabla G_m(\mathbf{X}[k]) + \sqrt{2\eta\tau}\mathbf{Z}[k], \quad \mathbf{X}[0] \sim \mu_0^{\otimes m} \tag{3}$$

where $\eta > 0$ is the step-size and $\mathbf{Z}[1], \mathbf{Z}[2], \ldots$ are i.i.d. standard Gaussian vectors (see Eq. (10) for an equivalent definition of NPGD directly in terms of $G$ and its first-variation).

When $G$ is linear, i.e. $G(\mu) = \int V \mathrm{d}\mu$ for some smooth $V : \mathbb{R}^d \to \mathbb{R}$, the particles $X_i$ are independent and each follows the (unadjusted) Langevin algorithm (Ermak, 1975; Roberts and Tweedie, 1996; Durmus and

Moulines, 2017) given by the stochastic recursion

$$X[k+1] = X[k] - \eta \nabla V(X[k]) + \sqrt{2\eta\tau}Z[k], \quad X[0] \sim \mu_0 \tag{4}$$

and it is thus sufficient to choose $m = 1$ in that case. In the general case of a convex and non-linear $G$, the particles will interact in non-trivial ways and $m$ should be taken large, so that a mean-field behavior emerges.

**Mean-Field Langevin**    The dynamics obtained in the many-particle $m \to \infty$ and vanishing step-size $\eta \to 0$ limit was called the *Mean-Field Langevin* dynamics in Hu et al. (2021) and is our object of interest. In this limit, the distribution $\mu_t$ of particles at time $t = k\eta$ solves the following drift-diffusion partial differential equation (PDE) of *McKean-Vlasov* type:

$$\partial_t \mu_t = \nabla \cdot \left( \mu_t \nabla V[\mu_t] \right) + \tau \Delta \mu_t \tag{5}$$

where $\nabla \cdot$ stands for the divergence operator and $V[\mu] \in \mathcal{C}^1(\mathbb{R}^d)$ is the *first-variation* of $G$ at $\mu$ (see Definition 2.1). This dynamics, which can be interpreted as the gradient flow of $F_\tau$ under the $W_2$ Wasserstein metric (Ambrosio and Savaré, 2007), is a generalization of the Langevin dynamics to a specific form of non-linear drift term.

There is a long line of work around mean-field dynamics (Dobrushin, 1979; Sznitman, 1991) (see Lacker (2018) for an introduction and references) which guarantee that NPGD (3) indeed converges to the Mean-Field Langevin dynamics, sometimes with fine quantitative bounds (Lacker, 2021; Mei et al., 2019). As for the behavior of the Mean-Field Langevin dynamics (5) itself, it is shown in (Mei et al., 2018; Hu et al., 2021) that, under suitable coercivity assumptions, $(\mu_t)$ weakly converges to the unique minimizer of $F_\tau$ as $t \to \infty$. Moreover, Hu et al. (2021) remarks that the results from Eberle et al. (2019) to obtain quantitative rates apply here, but this argument is restricted to the large noise regime and does not exploit the convexity of $G$. These works leave open the question of quantitative guarantees without a strong noise assumption.

## 1.1   Contributions and related work

Our contributions are the following:

- We prove that, under a certain uniform log-Sobolev inequality assumption (which is in particular satisfied in the settings of Mei et al. (2018); Hu et al. (2021)), solutions to (5) converge at a global exponential rate to the minimizer of $F_\tau$ (Theorem 3.2). The known convergence rate of the Langevin dynamics under a log-Sobolev inequality is recovered as a particular case when $G$ is linear.

- We study the annealed dynamics where the noise $\tau = \tau_t$ is time-dependent and decays as $\alpha/\log(t)$ and prove that for $\alpha > 0$ large enough, $G(\mu_t)$ converges towards the minimum of the *unregularized* functional $F_0 = G$ (Theorem 4.1).

- In Section 5, we show that our results apply to noisy gradient descent on infinitely wide two-layer neural networks and we provide numerical experiments for $G$ being a kernel Maximum Mean Discrepancy (MMD).

Let us mention that other algorithms to solve problems of the form (1) are possible. Nitanda et al. (2021) proposed a dual averaging scheme which involves a sequence of Langevin diffusions and enjoys a $O(1/t)$ convergence rate in the mean-field limit. For low-dimensional problems, one can resort to discretizing the measure on a fixed grid, which leads to a convex problem amenable to standard (Bregman) gradient descent algorithms (Tseng, 2010).

The long-time behavior of drift-diffusion PDEs of the form Eq. (5) has been studied in the mathematical physics literature. General convergence rates (under assumptions that imply a large noise in our context) are proved in Eberle et al. (2019). For interacting particule systems with an interaction kernel $k$ (discussed in Section 5.2), the case where $k(x, y) = h(x - y)$ with $h$ convex can be dealt with using the notion of *displacement convexity*, see (Villani, 2021, Chap. 9.6). In contrast, we rely on standard convexity, which corresponds to a positive semi-definite interaction kernel $k$.

Upon completion of this work, we became aware of the paper (Nitanda et al., 2022) which also proves the exponential convergence of the Mean-Field Langevin dynamics with the same proof technique. The main differences between these two works is that they perform a discrete time analysis while we study the annealed dynamics. These works were conducted independently and simultaneously.

## 1.2 Notations

We use $\| \cdot \|$ for the Euclidean norm on $\mathbb{R}^d$. For $\mu, \nu \in \mathcal{P}_2(\mathbb{R}^d)$, $\Pi(\mu, \nu)$ is the set of transport plans, that is, probability measures on $\mathbb{R}^d \times \mathbb{R}^d$ with marginals $\mu$ and $\nu$ respectively. The Wasserstein distance $W_2 : \mathcal{P}_2(\mathbb{R}^d)^2 \to \mathbb{R}_+$ is defined as the square-root of

$$W_2(\mu, \nu)^2 := \min_{\gamma \in \Pi(\mu, \nu)} \int_{(\mathbb{R}^d)^2} \|y - x\|^2 \mathrm{d}\gamma(x, y). \tag{6}$$

Relevant background on the Wasserstein distance can be found in Ambrosio and Savaré (2007). We often identify absolutely continuous probability measures with their density with respect to the Lebesgue measure. A function $f : \mathbb{R}^d \to \mathbb{R}$ is said $L$-smooth if its gradient is a $L$-Lipschitz continuous function.

## 2 Assumptions and preliminaries

### 2.1 First-variation and smoothness of $G$

The Mean-Field Langevin dynamics in Eq. (5) involves the *first-variation $V$* of $G$, defined as follows.

**Definition 2.1** (First-variation)**.** *We say that $G : \mathcal{P}_2(\mathbb{R}^d) \to \mathbb{R}$ admits a* first-variation *at $\mu \in \mathcal{P}_2(\mathbb{R}^d)$ if there exists a continuous function $V[\mu] : \mathbb{R}^d \to \mathbb{R}$ such that*

$$\forall \nu \in \mathcal{P}_2(\mathbb{R}^d), \ \lim_{\epsilon \downarrow 0} \frac{1}{\epsilon} \big( G((1 - \epsilon)\mu + \epsilon \nu) - G(\mu) \big) = \int_{\mathbb{R}^d} V[\mu](x) \mathrm{d}(\nu - \mu)(x). \tag{7}$$

*If it exists, the first-variation $V[\mu]$ is unique up to an additive constant.*

The notion of first-variation appears naturally when studying variational problems over $\mathcal{P}_2(\mathbb{R}^d)$ and its precise definition varies across references, see e.g. (Santambrogio, 2015, Def. 7.12). Throughout our work, we make the following regularity assumptions on $G$.

> **Assumption 1** (Smoothness of $G$)**.** *For all $\mu \in \mathcal{P}_2(\mathbb{R}^d)$, $G$ admits a first-variation $V[\mu] \in \mathcal{C}^1(\mathbb{R}^d)$ and $(\mu, x) \to \nabla V[\mu](x)$ is Lipschitz continuous in the following sense: there exists $L > 0$ such that*
>
> $$\forall \mu, \nu \in \mathcal{P}_2(\mathbb{R}^d), \quad \forall x, y \in \mathbb{R}^d, \quad \|\nabla V[\mu](x) - \nabla V[\nu](y)\|_2 \leq L \big( \|x - y\|_2 + W_2(\mu, \nu) \big).$$

Let us now state a lemma that is useful in our proofs, that gives the evolution of $G$ and $H$ along dynamics $(\mu_t)_{t \in (a,b)}$ in $\mathcal{P}_2^a(\mathbb{R}^d)$ that solve the *continuity equation* (in the sense of distributions):

$$\partial_t \mu_t = -\nabla \cdot (\mu_t v_t) \tag{8}$$

for some time-dependent velocity field $v \in L^2((a, b), L^2(\mu_t))$. Observe that Eq. (5) is an equation of this form with $v_t = -\nabla V[\mu_t] - \tau \nabla \log(\mu_t)$.

**Lemma 2.2** (Chain rule)**.** *Let $(\mu_t)_{t \in (a,b)}$ be a weakly continuous solution to Eq. (8) such that $\nabla \log(\mu_t) \in L^2((a,b), L^2(\mu_t))$. Then $G(\mu_t)$ and $H(\mu_t)$ are absolutely continuous functions of $t$ and it holds for a.e. $t \in (a, b)$,*

$$\frac{\mathrm{d}}{\mathrm{d}t} G(\mu_t) = \int_{\mathbb{R}^d} \nabla V[\mu_t]^\top v_t \mathrm{d}\mu_t \qquad and \qquad \frac{\mathrm{d}}{\mathrm{d}t} H(\mu_t) = \int_{\mathbb{R}^d} (\nabla \log(\mu_t))^\top v_t \mathrm{d}\mu_t.$$

*Proof.* Using the vocabulary of analysis in Wasserstein space, the function $H$ is displacement convex with subdifferential $\nabla \log \mu$ (Ambrosio and Savaré, 2007, Thm. 4.16). Also we prove in Lemma A.2 that $G$ is $(-2L)$-displacement convex with subdifferential $\nabla V[\mu_t]$. Then the claim is a consequence of (Ambrosio and Savaré, 2007, Sec. 4.4.E). □

## 2.2 Characterization of the minimizer

We recall the optimality conditions for $F_\tau$ which have been proved in several works (see e.g. (Mei et al., 2018, Lem. 10.4) or (Hu et al., 2021, Prop. 2.5)) and require the following assumptions.

> **Assumption 2.** *The function $G$ is convex and $F_\tau = G + \tau H$ admits a minimizer $\mu_\tau^*$.*

We stress that by convexity we mean standard convexity for the linear structure in $\mathcal{P}_2(\mathbb{R}^d)$, i.e.

$$\forall \mu, \nu \in \mathcal{P}_2(\mathbb{R}^d), \forall \alpha \in [0,1], \quad G(\alpha\mu + (1-\alpha)\nu) \le \alpha G(\mu) + (1-\alpha)G(\nu).$$

**Proposition 2.3.** *Under Assumption 1 and 2, the minimizer $\mu_\tau^*$ of $F$ is unique and satisfies*

$$\mu_\tau^* \propto e^{-V[\mu_\tau^*]/\tau}. \tag{9}$$

The uniqueness comes from the strict convexity of $H$. For Eq. (9), one first derives the first order optimality condition, which require that $V[\mu_\tau^*] + \tau \log(\mu_\tau^*)$ must be a constant $\mu_\tau^*$-almost everywhere. Then one shows that $\mu_\tau^*$ has positive density everywhere due to the entropy term, and concludes. We refer to (Mei et al., 2018, Lem. 10.4) for details.

## 2.3 Noisy Particle Gradient Descent (NPGD)

The NPGD algorithm has been defined in Section 1 via the function $G_m$. We now give an alternative definition of this algorithm involving the first-variation $V$ of $G$.

Assume that $G$ satisfies Assumption 1, let $V$ be its first-variation and fix $m \in \mathbb{N}^*$. For $i \in [m]$, initialize randomly $X_{i,0} \overset{iid}{\sim} \mu_0 \in \mathcal{P}_2(\mathbb{R}^d)$ and define recursively for $k \ge 0$

$$\begin{cases} X_{i,k+1} = X_{i,k} - \eta \nabla V[\hat{\mu}_k](X_{i,k}) + \sqrt{2\eta\tau} Z_{i,k} \\ \hat{\mu}_k = \dfrac{1}{m} \sum_{i=1}^m \delta_{X_{i,k}} \end{cases} \tag{10}$$

where $\eta > 0$ is the step-size and $Z_{i,k} \sim \mathcal{N}(0, I)$ are iid standard Gaussian random variables.

**Proposition 2.4.** *Under Assumption (1), the two definitions of NPGD in Eq. (3) and in Eq. (10) are equivalent.*

*Proof.* For $\mathbf{X} \in (\mathbb{R}^d)^m$ and $\mathbf{Y} \in (\mathbb{R}^d)^m$ define $\mu_t = \frac{1}{m} \sum_{i=1}^m \delta_{X_i + tY_i}$. It satisfies the continuity equation (8) with velocity field $v_t(X_i + tY_i) = Y_i$. By Lemma 2.2, it holds

$$\frac{\mathrm{d}}{\mathrm{d}t} G_m(\mathbf{X} + t\mathbf{Y})|_{t=0} = \frac{\mathrm{d}}{\mathrm{d}t} G(\mu_t)|_{t=0} = \int_{\mathbb{R}^d} \nabla V[\mu_0](x)^\top v_0(x) \mathrm{d}\mu_0(x) = \frac{1}{m} \sum_{i=1}^m \nabla V[\mu_0](X_i)^\top Y_i.$$

This proves that $\forall i \in [m]$, $m\nabla_{X_i} G_m(\mathbf{X}) = \nabla V[\mu_0](X_i)$ and thus the update equations in Eq. (3) and Eq. (10) are the same. $\qquad\square$

## 2.4 Mean-Field Langevin dynamics

Given Eq. (10) standard results about mean-field systems tell us that as $m \to \infty$, the random measure $\hat{\mu}_k$ becomes deterministic, so that in the limit (and taking also the small-step size limit $\eta \to 0$) the particles trajectories are given by i.i.d. samples from the following stochastic differential equation (SDE)

$$\begin{cases} \mathrm{d}X_t = -\nabla V[\mu_t](X_t)\mathrm{d}t + \sqrt{2\tau}\mathrm{d}B_t, \quad X_0 \sim \mu_0 \\ \mu_t = \mathrm{Law}(X_t) \end{cases} \tag{11}$$

where $(B_t)_{t\geq 0}$ is a Brownian motion. As mentioned in the introduction, the law $(\mu_t)$ of a solution to this SDE solves the following PDE which is our main object of study:

$$\partial_t \mu_t = \nabla \cdot \big(\mu_t \nabla V[\mu_t]\big) + \tau \Delta \mu_t \tag{12}$$

where $\nabla\cdot$ stands for the divergence operator. Standard results about this class of PDEs guarantee its well-posedness, i.e. the existence of a unique solution, under Assumption 1 (see e.g. (Huang et al., 2021, Thm. 3.3) or Ambrosio and Savaré (2007) for an approach based on the gradient flow structure which applies here thanks to Lemma A.2 which states that $G$ is $(-2L)$-displacement convex).

Let us now study the convergence of $(\mu_t)_{t\geq 0}$ to the global minimum of $F_\tau$.

## 3 Exponential convergence of Mean-Field Langevin dynamics

For $\mu, \nu \in \mathcal{P}_2^a(\mathbb{R}^d)$ with $\mu$ absolutely continuous w.r.t. $\nu$ we define the *relative entropy* (a.k.a. Kullback-Leibler divergence) by

$$H(\mu|\nu) \coloneqq \int_{\mathbb{R}^d} \log\Big(\frac{\mathrm{d}\mu}{\mathrm{d}\nu}\Big)\mathrm{d}\mu,$$

and the *relative Fisher information* by

$$I(\mu|\nu) \coloneqq \int_{\mathbb{R}^d} \Big\|\nabla \log \frac{\mathrm{d}\mu}{\mathrm{d}\nu}\Big\|^2 \mathrm{d}\mu.$$

**Definition 3.1** (Log-Sobolev inequality)**.** *We say that $\nu \in \mathcal{P}_2^a(\mathbb{R}^d)$ satisfies a logarithmic Sobolev inequality with constant $\rho > 0$ (in short $\mathrm{LSI}(\rho)$) if for all $\mu \in \mathcal{P}_2^a(\mathbb{R}^d)$ absolutely continuous w.r.t. $\nu$, it holds*

$$H(\mu|\nu) \leq \frac{1}{2\rho}I(\mu|\nu). \tag{13}$$

This inequality can be interpreted as a 2-Łojasiewicz gradient inequality for the functional $\mu \mapsto H(\mu|\nu) = \int V \mathrm{d}\mu + H(\mu)$ (where we have posed $V = -\log\nu$) in the Wasserstein geometry (Otto and Villani, 2000) and thus directly implies the exponential convergence of its Wasserstein gradient flow. This corresponds to our objective function in the linear case $G(\mu) = \int V\mathrm{d}\mu$, and in this case exponential convergence towards minimizers is thus guaranteed when $\nu = e^{-V}$ satisfies a Log-Sobolev inequality.

In the general case, we make an analogous assumption that such an inequality holds *uniformly* for $e^{-V[\mu_t]/\tau}$ throughout the dynamics.

**Assumption 3** (Uniform log-Sobolev)**.** *There exists $\rho_\tau > 0$ such that $\forall \mu \in \mathcal{P}_2(\mathbb{R}^d)$ it holds $e^{-V[\mu]/\tau} \in L^1(\mathbb{R}^d)$ and the probability measure $\nu \propto e^{-V[\mu]/\tau}$ satisfies $\mathrm{LSI}(\rho_\tau)$.*

Remember that $V[\mu]$ is defined up to a constant term, and in this section, we fix this constant so that $e^{-V[\mu]/\tau} \in \mathcal{P}_2(\mathbb{R}^d)$. Let us recall two criteria for a probability measure to satisfy a Log-Sobolev inequality:

– If $\nabla^2 V \succeq \rho I_d$ then $e^{-V} \in \mathcal{P}(\mathbb{R}^d)$ satisfies $\mathrm{LSI}(\rho)$ (Bakry and Émery, 1985);

– if $\nu$ satisfies $\mathrm{LSI}(\rho)$ and $\tilde{\nu} = e^{-\psi}\nu \in \mathcal{P}(\mathbb{R}^d)$ is a perturbation of $\nu$ with $\psi \in L^\infty(\mathbb{R}^d)$ then $\tilde{\nu}$ satisfies $\mathrm{LSI}(\tilde{\rho})$ with $\tilde{\rho} = \rho e^{\inf\psi - \sup\psi}$ (Holley and Stroock, 1987).

These two criteria are standard, but many finer criteria are known, such as integral conditions (Wang, 2001), Lyapunov conditions (Cattiaux et al., 2010) or criteria for mixture distributions (Chen et al., 2021) (see also Section 5.1). Our main result regarding the Mean-Field Langevin dynamics (11) is the following.

**Theorem 3.2.** *Under Assumptions 1, 2 and 3, let $\mu_0 \in \mathcal{P}_2(\mathbb{R}^d)$ be such that $F_\tau(\mu_0) < \infty$. For $t \geq 0$, it holds*

$$F_\tau(\mu_t) - F_\tau(\mu_\tau^*) \leq e^{-2\tau\rho_\tau t}(F_\tau(\mu_0) - F_\tau(\mu_\tau^*)). \tag{14}$$

*Proof.* Let $\nu_t = e^{-V[\mu_t]/\tau}$. By Lemma 2.2 applied to $v_t = -\nabla V[\mu_t] - \tau \nabla \log(\mu_t)$, we have

$$\frac{d}{dt}F_\tau(\mu_t) = -\int_{\mathbb{R}^d} \|\nabla V[\mu_t] + \tau \nabla \log(\mu_t)\|^2 \, d\mu_t = -\tau^2 I(\mu_t|\nu_t). \tag{15}$$

Note that although Lemma 2.2 requires some regularity estimates, they can be bypassed here thanks to general results about Wasserstein gradient flows (Ambrosio and Savaré, 2007, Thm. 5.3 (v)). Combining this energy identity with the log-Sobolev inequality and Lemma 3.4, it follows

$$\frac{d}{dt}\big(F_\tau(\mu_t) - F_\tau(\mu_\tau^*)\big) = -\tau^2 I(\mu_t|\nu_t) \leq -2\rho_\tau \tau^2 H(\mu_t|\nu_t) \leq -2\rho_\tau \tau(F(\mu_t) - F(\mu^*))$$

which is a 2-Łojasiewicz gradient inequality for $F_\tau$. By integrating in time we get Eq. (14). $\qquad\square$

In the proof, we see that we could relax Assumption 3 and require the Log-Sobolev inequality to hold only for all $\mu \in \mathcal{P}_2(\mathbb{R}^d)$ such that $F_\tau(\mu) \leq F_\tau(\mu_0)$. Also, Assumption 1 is only a general assumption that guarantees well-posedness of the dynamics and the energy decay formula Eq. (15); this regularity assumption can be relaxed on a case by case basis. Convergence guarantees in parameter space directly follow from the previous theorem.

**Corollary 3.3.** *Under the assumptions of Theorem 3.2, for $t \geq 0$ we have*

$$H(\mu_t|\mu_\tau^*) \leq \frac{1}{\tau}e^{-2\tau\rho_\tau t}(F_\tau(\mu_0) - F_\tau(\mu^*)) \qquad and \qquad W_2^2(\mu_t, \mu_\tau^*) \leq \frac{2e^{-2\tau\rho_\tau t}}{\tau\rho_\tau}\big(F_\tau(\mu_0) - F_\tau(\mu^*)\big).$$

*Proof.* The first inequality follows from Theorem 3.2 and Lemma 3.4. For the second one, it follows from the fact that if $\nu$ satisfies LSI($\rho$), then it satisfies the *Talagrand inequality*, which states that $\forall \mu \in \mathcal{P}_2(\mathbb{R}^d)$, $W_2^2(\mu, \nu) \leq \frac{2}{\rho}H(\mu|\nu)$, as proved in Otto and Villani (2000). $\qquad\square$

The following lemma establishes inequalities which are key to handle the non-linear aspect of the dynamics (when $G$ is linear, they become trivial equalities).

**Lemma 3.4** (Entropy Sandwich). *Under Assumption 1, 2 and 3, let $\mu_\tau^*$ be the unique minimizer of $F_\tau$. For all $\mu \in \mathcal{P}_2(\mathbb{R}^d)$, letting $\nu := e^{-V[\mu]/\tau} \in \mathcal{P}_2(\mathbb{R}^d)$, it holds*

$$\tau H(\mu|\mu_\tau^*) \leq F_\tau(\mu) - F_\tau(\mu_\tau^*) \leq \tau H(\mu|\nu).$$

*Proof.* The convexity of $G$ implies that, $\forall \mu, \nu \in \mathcal{P}_2(\mathbb{R}^d)$, $\frac{1}{\epsilon}(G((1-\epsilon)\mu + \epsilon\nu) - G(\mu)) \leq G(\nu) - G(\mu)$. So, passing to the limit in the definition of the first-variation (Definition 2.1), we recover the usual convexity inequality (interpreting $V[\mu]$ as the gradient of $G$ at $\mu$):

$$G(\nu) \geq G(\mu) + \int V[\mu]d(\nu - \mu). \tag{16}$$

Invoking this inequality twice with the role of $\mu$ and $\mu^*$ exchanged, it holds

$$\int V[\mu^*]d(\mu - \mu^*) \leq G(\mu) - G(\mu^*) \leq \int V[\mu]d(\mu - \mu^*).$$

Recalling $F_\tau(\mu) = G(\mu) + \tau H(\mu)$, it holds, on the one hand,

$$F_\tau(\mu) - F_\tau(\mu^*) \leq \int V[\mu]d\mu + \tau H(\mu) - \int V[\mu]d\mu^* - \tau H(\mu^*)$$
$$= \tau H(\mu|\nu) - \tau H(\mu^*|\nu) \leq \tau H(\mu|\nu).$$

On the other hand, using the fact that $\mu_\tau^* = e^{-V[\mu^*]/\tau}$ (Proposition 2.3), it holds

$$F_\tau(\mu) - F_\tau(\mu_\tau^*) \geq \int V[\mu^*]\mathrm{d}\mu + \tau H(\mu) - \int V[\mu^*]\mathrm{d}\mu^* - \tau H(\mu^*)$$

$$= \tau H(\mu|\mu^*) - \tau H(\mu^*|\mu^*) = \tau H(\mu|\mu^*). \qquad \square$$

## 4 Convergence of the annealed dynamics

We now turn our attention to the "annealed" Mean-Field Langevin dynamics

$$\partial \mu_t = \nabla \cdot \left( \mu_t \nabla V[\mu_t] \right) + \tau_t \Delta \mu_t \tag{17}$$

with a time-dependent *temperature* parameter $\tau_t$ that converges to 0. The existence of a unique solution from any $\mu_0 \in \mathcal{P}_2(\mathbb{R}^d)$ follows again from the theory of McKean-Vlasov equations, now with time inhomogeneous coefficients (see e.g. (Huang et al., 2021, Thm. 3.3)). As a side note, notice that (17) cannot strictly be interpreted as a Wasserstein gradient flow anymore, but some aspects of the theory of Wasserstein gradient flows have been extended to cover the case of time-dependent diffusion coefficients (Ferreira and Valencia-Guevara, 2018, Sec. 6.2).

The linear case when $G(\mu) = \int V \mathrm{d}\mu$ has been considered in numerous works (e.g. Holley et al. (1989); Geman and Hwang (1986); Miclo (1992); Raginsky et al. (2017); Tang and Zhou (2021)). It is known in particular (Miclo, 1992) that under suitable coercivity assumptions for $V$ and if $\tau_t = C/\log(t)$ for some $C > 0$ large enough, then $G(\mu_t)$ converges to $\min_{\mu \in \mathcal{P}_2(\mathbb{R}^d)} G(\mu) = \min_{x \in \mathbb{R}^d} V(x)$.

Here we show that a similar guarantee holds in our more general context.

**Theorem 4.1** (Convergence of annealed dynamics). *Suppose Assumptions 1, 2 and 3 hold for all $\tau > 0$, and moreover assume that:*

- *the Log-Sobolev constants satisfy $\rho_\tau \geq C_0 e^{-\alpha^*/\tau}$ for some $\alpha^*, C_0 > 0$,*

- *$G$ is lower-bounded,*

- *$(\tau_t)_t$ is smooth, decreases, and for $t$ large it holds $\tau_t = \alpha/\log(t)$ for some $\alpha > \alpha^*$.*

*Let $\mu_0 \in \mathcal{P}_2(\mathbb{R}^d)$ be such that $F_{\tau_0}(\mu_0) < \infty$. Then for each $\epsilon > 0$, there exists $C, C' > 0$ such that*

$$F_{\tau_t}(\mu_t) - F_{\tau_t}(\mu_{\tau_t}^*) \leq C t^{-\left(1 - \frac{\alpha^*}{\alpha} - \epsilon\right)}, \tag{18}$$

*and*

$$G(\mu_t) - \inf G \leq C' \frac{\log \log t}{\log t}. \tag{19}$$

We can make the following comments:

- The lower-bound assumed on $\rho_\tau$ is natural when one has in mind the Holley and Stroock criterion given in Section 3. In Section 5, we show a lower bound of this form on a concrete example related to two-layer neural networks.

- The bounds of Theorem 4.1 exhibit a two time-scales phenomenon: the dynamics $(\mu_t)$ converges at a polynomial rate to the regularization path $(\mu_{\tau_t}^*)$ (in relative entropy or $W_2^2$ distance, thanks to the "entropy sandwich" Lemma 3.4 or the Talagrand inequality) but the regularization path only converges at a logarithmic rate to the optimal value $\inf G$, because of the slow decay of $\tau_t$.

- The slow decay of $\tau_t$ is an inconvenience but it cannot be improved. It is known that in the linear case $G(\mu) = \int V \mathrm{d}\mu$, convergence is lost if $\tau_t$ decays faster (Holley et al., 1989, Sec. 3) (in fact, taking $\tau_t = \alpha/\log(t)$ with $\alpha > 0$ too small already breaks convergence).

*Proof.* Our proof is partly inspired by Miclo (1992), as revisited by Tang and Zhou (2021).

**Step 1.** Consider the function that returns the values of the regularization path

$$h(\tau) \coloneqq F_\tau(\mu_\tau^*) = \min_{\mu \in \mathcal{P}_2(\mathbb{R}^d)} G(\mu) + \tau H(\mu).$$

As an infimum of affine functions, $h$ is concave and since the minimizer $\mu_\tau^*$ is unique, $h$ is differentiable for $\tau > 0$ and its derivative is $h'(\tau) = H(\mu_\tau^*)$. We focus on $t \geq t_0$ so that $\tau_t = \alpha/\log(t)$. By Lemma 2.2 applied to $v_t = -\nabla V[\mu_t] + \tau_t \nabla \log(\mu_t)$ (here again, the regularity assumptions of Lemma 2.2 can be bypassed using the gradient flow-like structure, see (Ferreira and Valencia-Guevara, 2018, Thm. 6.9)), we have

$$\frac{d}{dt}\big(F_{\tau_t}(\mu_t) - F_{\tau_t}(\mu_{\tau_t}^*)\big) = -\int \|\nabla V[\mu_t] + \tau_t \nabla \log(\mu_t)\|^2 d\mu_t + \tau_t' H(\mu_t) - \tau_t' h'(\tau_t)$$

$$\leq -\tau_t^2 I(\mu_t|\nu_t) + \tau_t'\big(H(\mu_t) - H(\mu_{\tau_t}^*)\big)$$

where we introduced the probability measure $\nu_t \propto e^{-V[\mu_t]/\tau_t}$. On the one hand, we have by the Log-Sobolev inequality and the "entropy sandwich" Lemma 3.4,

$$\tau_t^2 I(\mu_t|\nu_t) \geq 2\tau_t^2 \rho_{\tau_t} H(\mu_t|\nu_t) \geq 2\rho_{\tau_t}\tau_t\big(F_{\tau_t}(\mu_t) - F_{\tau_t}(\mu_{\tau_t}^*)\big).$$

On the other hand, by Lemma 4.2 below, it holds for some $C_2, C_3 > 0$ independent from $\mu$ and $t$,

$$-\tau_t(H(\mu_t) - H(\mu_{\tau_t}^*)) \leq C_2 \tau_t F_{\tau_t}(\mu_t) + \tau_t C_3 + F_{\tau_t}(\mu_{\tau_t}^*) - G(\mu_{\tau_t}^*)$$

$$\leq C_2 \tau_t (F_{\tau_t}(\mu_t) - F_{\tau_t}(\mu_{\tau_t}^*)) + (1 + C_2 \tau_t) F_{\tau_t}(\mu_{\tau_t}^*) - G(\mu_{\tau_t}^*) + \tau_t C_3$$

$$\leq C_2'(F_{\tau_t}(\mu_t) - F_{\tau_t}(\mu_{\tau_t}^*)) + C_3'$$

where in the last step, we used that $G$ is lower bounded and $h(\tau) = F_\tau(\mu_\tau^*)$ is bounded for $\tau \in [0, \tau_0]$. Combining the previous estimates, we get that for any $\epsilon > 0$, there exists $C_1, C_2, C_3 > 0$ such that

$$\frac{d}{dt}\big(F_{\tau_t}(\mu_t) - F_{\tau_t}(\mu_{\tau_t}^*)\big) \leq -2\rho_{\tau_t}\tau_t\big(F_{\tau_t}(\mu_t) - F_{\tau_t}(\mu_{\tau_t}^*)\big) - C_2\frac{\tau_t'}{\tau_t}\big(F_{\tau_t}(\mu_t) - F_{\tau_t}(\mu_{\tau_t}^*)\big) - \frac{\tau_t'}{\tau_t}C_3$$

$$\leq \frac{-2C_1\alpha t^{-\frac{\alpha^*}{\alpha}} + C_2 t^{-1}}{\log t}\big(F_{\tau_t}(\mu_t) - F_{\tau_t}(\mu_{\tau_t}^*)\big) + C_3\frac{t^{-1}}{\log t}$$

where we used $\tau_t = \alpha/\log(t)$, $\tau_t' = -\alpha/(t(\log t)^2)$ and $\rho_{\tau_t} \geq C_0 t^{-\alpha^*/\alpha}$. In passing, the first inequality in the above display guarantees that $F_{\tau_t}(\mu_t) - F_{\tau_t}(\mu_{\tau_t}^*)$ remains finite at all time because $\log \tau_t \in \mathcal{C}^1$, which justifies the fact that we can consider only $t$ large enough in the rest of the proof.

It follows that for any $\epsilon > 0$ such that $\epsilon < 1 - \alpha^*/\alpha$, for $t$ large enough and some $C, C' > 0$,

$$\frac{d}{dt}\big(F_{\tau_t}(\mu_t) - F_{\tau_t}(\mu_{\tau_t}^*)\big) \leq -Ct^{-\frac{\alpha^*}{\alpha}-\epsilon}\big(F_{\tau_t}(\mu_t) - F_{\tau_t}(\mu_{\tau_t}^*)\big) + C't^{-1}/\log(t).$$

Now define

$$Q(t) \coloneqq \big(F_{\tau_t}(\mu_t) - F_{\tau_t}(\mu_{\tau_t}^*)\big) - \frac{C'}{C}t^{-1+\frac{\alpha^*}{\alpha}+\epsilon}$$

which satisfies

$$\frac{d}{dt}Q(t) \leq -Ct^{-\frac{\alpha^*}{\alpha}-\epsilon}Q(t) - C't^{-1} + Ct^{-1}/\log(t) + \frac{C'(1-\frac{\alpha^*}{\alpha}-\epsilon)}{C}t^{-2+\frac{\alpha^*}{\alpha}+\epsilon}. \tag{20}$$

Observe that the term $-C't^{-1}$ dominates the two last terms for $t$ large enough. Thus for $t \geq t_*$ large enough, $\frac{d}{dt}Q(t) \leq -Ct^{-\frac{\alpha^*}{\alpha}-\epsilon}Q(t)$ which implies $Q(t) \leq Q(t_*)\exp(-C\int_{t_*}^t s^{-\frac{\alpha^*}{\alpha}-\epsilon}ds)$. As a consequence

$$F_{\tau_t}(\mu_t) - F_{\tau_t}(\mu_{\tau_t}^*) \leq \frac{C'}{C}t^{-1+\frac{\alpha^*}{\alpha}+\epsilon} + Q(t_*)\exp\Big(-\frac{C}{\kappa}(t^\kappa - t_*^\kappa)\Big)$$

and thus $F_{\tau_t}(\mu_t) - F_{\tau_t}(\mu_{\tau_t}^*) \leq C'' t^{-\kappa}$ because $\kappa := 1 - \frac{\alpha^*}{\alpha} - \epsilon > 0$ and $Q(t^*)$ is finite. This proves Eq. (18).

**Step 2.** Let us now prove Eq. (19), under the assumption that $G$ admits a minimizer $\mu_0^* \in \mathcal{P}_2(\mathbb{R}^d)$. The proof can be easily adapted to the general case by choosing $\mu_0^*$ as a quasi-minimizer such that $G(\mu_0^*) \leq \inf G + \epsilon$ and taking $\epsilon$ arbitrarily small. Remember that $h(0) = G(\mu_0^*) = F_0(\mu_0^*)$, so

$$
\begin{aligned}
G(\mu_t) - G(\mu_0^*) &= F_{\tau_t}(\mu_t) - F_{\tau_t}(\mu_{\tau_t}^*) + F_{\tau_t}(\mu_{\tau_t}^*) - F_0(\mu_0^*) - \tau_t H(\mu_t) \\
&\leq C t^{-\kappa} + (h(\tau_t) - h(0)) + C' \tau_t
\end{aligned}
$$

where we have used the bound $-H(\mu_t) \leq C_1 F_{\tau_t}(\mu_t) + C_2$ from Lemma 4.2, which is uniformly bounded for $t \geq 0$ by some $C'$ thanks to Step 1.

The rest of the proof consists in bounding $h(\tau) - h(0)$ via an approximation argument. Let $g_\sigma(x) = (2\pi\sigma^2)^{-d/2} \exp(-\|x\|^2/(2\sigma^2))$ be the standard Gaussian kernel and let $\tilde{\mu}_\sigma(x) = \int g_\sigma(x-y)\mathrm{d}\mu_0^*(y)$. We consider the transport plan $\gamma \in \Pi(\tilde{\mu}_0, \tilde{\mu}_\sigma)$ given by the joint law of $(X, X+Z)$ for $\mathrm{Law}(X) = \tilde{\mu}_0 = \mu_0^*$ and $\mathrm{Law}(Z) = g_\sigma$. On the one hand, it holds by convexity of $G$

$$
\begin{aligned}
0 \geq G(\tilde{\mu}_0) - G(\tilde{\mu}_\sigma) &\geq \int V[\tilde{\mu}_\sigma]\mathrm{d}[\tilde{\mu}_0 - \tilde{\mu}_\sigma] \\
&= \int (V[\tilde{\mu}_\sigma](y) - V[\tilde{\mu}_\sigma](x))\mathrm{d}\gamma(x,y).
\end{aligned}
$$

It follows, using the smoothness bound $V[\tilde{\mu}_\sigma](x) - V[\tilde{\mu}_\sigma](y) \leq \nabla V[\tilde{\mu}_\sigma](y)^\top (x-y) + \frac{L}{2}\|y-x\|^2$ and the fact that the Gaussian kernel is centered, that

$$
\begin{aligned}
|G(\tilde{\mu}_0) - G(\tilde{\mu}_\sigma)| &\leq \int (V[\tilde{\mu}_\sigma](x) - V[\tilde{\mu}_\sigma](y))\mathrm{d}\gamma(x,y) \\
&\leq \int \nabla V[\tilde{\mu}_\sigma](x)^\top (y-x)\mathrm{d}\gamma(x,y) + \frac{L}{2}\int \|y-x\|^2\mathrm{d}\gamma(x,y) \\
&= 0 + \frac{L}{2}\sigma^2.
\end{aligned}
$$

On the other hand, we have by Jensen's inequality for the convex function $\varphi : s \mapsto s\log(s)$ and Fubini's theorem:

$$
\begin{aligned}
H(\tilde{\mu}_\sigma) &= \int \varphi\Big( \int g_\sigma(x-y)\mathrm{d}\mu_0(y)\Big)\mathrm{d}x \\
&\leq \int \Big( \int \varphi(g_\sigma(x-y))\mathrm{d}x\Big)\mathrm{d}\mu_0(y) = -\frac{1}{2}\big(1 + \log(2\pi\sigma^2)\big)
\end{aligned}
$$

which is the entropy of the Gaussian distribution $g_\sigma$. Thus we have

$$
h(\tau) - h(0) \leq \inf_{\sigma > 0} \frac{L}{2}\sigma^2 - \frac{\tau}{2}\big(1 + \log(2\pi\sigma^2)\big) \leq -\frac{\tau}{2}\log(\pi\tau)
$$

by choosing $\sigma^2 = \tau/L$. Plugging the value of $\tau_t = \alpha/\log(t)$ we get, for some $C, C' > 0$,

$$
G(\mu_t) - G(\mu_0^*) \leq \frac{\alpha}{2}\frac{(\log\log t - \log(\pi\alpha))}{\log t} + C t^{-\kappa} + C'\frac{\alpha}{\log(t)} \leq C''\frac{\log\log t}{\log t}. \qquad \square
$$

In the proof of Theorem 4.1, we used a lower bound on the value of $H(\mu)$ in terms of the functional value that is provided in the following lemma.

**Lemma 4.2.** *Under the assumptions of Theorem 4.1, there exists $C_1, C_2, C_3 > 0$ such that for all $0 < \tau \leq \tau_0$ and $\mu \in \mathcal{P}_2^a(\mathbb{R}^d)$, it holds $F_\tau(\mu) \geq -C_3$ and*

$$
-H(\mu) \leq C_1 F_\tau(\mu) + C_2.
$$

*Proof.* In the following proof, $C_i, C_i', C_i'' > 0$ are constants independent from $\mu$ which value may change from line to line. Since by assumption the probability measure $\nu$ proportional to $e^{-V[\mu_0]}$ satisfies a logarithmic Sobolev inequality, there exists $C_1, C_2 > 0$ such that $\forall x \in \mathbb{R}^d$, $V[\mu_0](x) \geq C_1 \|x\|^2 - C_2$. Indeed, by Herbst argument (Bakry et al., 2014, Prop. 5.4.1), there exists $C_1 > 0$ such that $\int e^{C_1\|x\|^2 - V[\mu_0](x)} dx < \infty$ and we conclude using the fact that if $f \in \mathcal{C}^1(\mathbb{R}^d)$ has a Lipschitz gradient and $\int e^f dx < \infty$ then $f$ must be upper-bounded.

Letting $M_2(\mu) \coloneqq \int \|x\|^2 d\mu(x)$, it follows, using convexity of $G$, that

$$G(\mu) \geq G(\mu_0) + \int V[\mu_0] d(\mu - \mu_0) \geq 2C_1 M_2(\mu) - C_2.$$

Invoking Lemma 4.3 with $\sigma^2 = \tau/C_1$ we have

$$\tau H(\mu) \geq -C_1 M_2(\mu) - \tau - \tau d \log(2\tau\pi/C_1).$$

Summing the two previous equations (with the same value of $C_1$), we get that for $\tau \leq \tau_0$,

$$F_\tau(\mu) \geq C_1 M_2(\mu) - C_2'.$$

Combined with the fact that $-H(\mu) \leq C_1' M_2(\mu) + C_2'$, we get $-H(\mu) \leq C_1'' F_\tau(\mu) + C_2''$. $\qquad\square$

See e.g. (Mei et al., 2018, Lem. 10.1) for a proof of the following lemma.

**Lemma 4.3.** *For $\mu \in \mathcal{P}_2^a(\mathbb{R}^d)$, let $M_2(\mu) \coloneqq \int \|x\|^2 d\mu(x)$. For any $\sigma^2 > 0$, it holds*

$$-H(\mu) \leq \frac{1}{\sigma^2} M_2(\mu) + 1 + d \log(2\pi\sigma^2).$$

# 5 Applications and experiments

## 5.1 Noisy GD on a wide two-layer neural network

We now show that our results apply to the training dynamics of certain wide 2-layer neural networks trained with noisy gradient descent.

Let us introduce the formulation of two-neural networks of arbitrary width parameterized by a probability measure, which is at the heart of the mean-field analysis of the training dynamics (Nitanda and Suzuki, 2017; Mei et al., 2018; Sirignano and Spiliopoulos, 2020; Rotskoff and Vanden-Eijnden, 2018; Chizat and Bach, 2018). Consider a input/output data distribution $(z, y) \sim \mathcal{D} \in \mathcal{P}(\mathbb{R}^n \times \mathbb{R})$, a loss function $\ell : \mathbb{R}^2 \to \mathbb{R}_+$, a "feature function" $\Phi(z, x) \in \mathcal{C}(\mathbb{R}^n \times \mathbb{R}^d)$ and let

$$G(\mu) \coloneqq \mathbf{E}_{(z,y)\sim\mathcal{D}} \, \ell\Big(y, \int \Phi(z, x) d\mu(x)\Big) + \frac{\lambda}{2} \int \|x\|^2 d\mu(x) \tag{21}$$

where $\lambda > 0$ is regularization parameter. Typical choices for the loss are the logistic loss $\ell(y, y') = \log(1 + \exp(-yy'))$ and the square loss $\ell(y, y') = \frac{1}{2}|y - y'|^2$ and in what follows, $\ell'$ denotes the derivative of $\ell$ with respect to $y'$.

When $\mu = \frac{1}{m} \sum_{i=1}^m \delta_{x_i}$ is an empirical distribution with $m$ atoms/particles, the function $G_m$ derived from $G$ as in Eq. (2) is exactly the risk with weight decay regularization for a two-layer neural network of width $m$. Thus noisy gradient descent for two-layer neural networks is equivalent to NPGD with $G$ defined in Eq (21).

Let us give simple conditions under which our convergence theorems apply in this case.

**Proposition 5.1.** *Assume that $\ell$ is the square or the logistic loss, that $|\Phi|$ is bounded by $K > 0$ and that $\Phi$ smooth in $x$, uniformly in $z$. Then Assumptions 1 and 2 are satisfied and the first variation of $G$ is given, for $\mu \in \mathcal{P}_2(\mathbb{R}^d)$ and $x \in \mathbb{R}^d$, by*

$$V[\mu](x) = \mathbf{E}_{(z,y)\sim\mathcal{D}} \, \ell'\Big(y, \int \Phi(z, x') d\mu(x')\Big) \Phi(z, x) + \frac{\lambda}{2} \|x\|^2.$$

*Moreover Assumption 3 is satisfied when:*

- $\ell$ is the logistic loss. Then we have $\rho_\tau \geq \frac{\lambda}{\tau} e^{-2K/\tau}$, or

- $\ell$ is the square loss and $|\mathbf{E}[y|z]| \leq K'$ a.s. Then we have $\rho_\tau \geq \frac{\lambda}{\tau} e^{-2K(K+K')/\tau}$.

*Proof.* For the computation of the first variation and Assumptions 1, we refer e.g. to Hu et al. (2021). For Assumption 2, $G$ is convex as a composition of a linear operator and a convex function. To see that $F_\tau$ admits a minimizer, notice that thanks to the regularization term, the sublevel sets of $G$ are tight and thus weakly-precompact by Prokorov's theorem. Moreover, the loss term in $G$ is weakly continuous, the regularization term is weakly lower-semicontinuous (lsc) and $H$ is weakly lsc (Ambrosio and Savaré, 2007, Sec. 3.2) so, overall, $F_\tau$ is lsc. Thus a minimizer $\mu_\tau^*$ exists for all $\tau \geq 0$ by the Direct Method in the calculus of variations.

Let us derive the lower-bound on the log-Sobolev constant $\rho_\tau$ using the criteria given below Assumption 3. First, by the Bakry-Émery criterion, the probability measure $\propto e^{-\frac{\lambda}{2\tau}\|x\|^2}$ satisfies LSI($\lambda/\tau$). Also, our assumptions guarantee that the first term in $V$ is uniformly bounded by $K$ – in case of the logistic loss because $|\ell'(y, y')| \leq 1$ – or by $K(K + K')$ – in case of the square loss. We conclude by applying the Holley-Stroock criterion with a perturbation $\psi$ that satisfies $\sup \psi - \inf \psi \leq 2K/\tau$ (for the logistic loss) or $\sup \psi - \inf \psi \leq 2K(K + K')/\tau$ (for the square loss). $\square$

**Limitations of this approach**   While the previous proposition, combined with our theorems, gives new convergence guarantees for noisy gradient descent on neural networks (in a certain limit), let us stress on the limitations of these results. First, the convergence proof fundamentally relies on the existence of noise and our analysis misses the fact that, in certain contexts, $G$ has a specific structure (of a different nature than the uniform LSI) that in practice seems to ease convergence and even make the noiseless dynamics converge, see e.g. Bach and Chizat (2021). Second, this approach introduces, in addition to weight decay, an entropic regularization which might be detrimental to the statistical performance. Finally, the risk for a vanilla two-layer neural network with non-linearity $\phi : \mathbb{R} \to \mathbb{R}$ is obtained from Eq. (21) by taking

$$\Phi(z, x) = a\phi(b^\top z) \quad \text{where} \quad x = (a, b) \in \mathbb{R} \times \mathbb{R}^{d-1} \tag{22}$$

which is not covered by our assumptions (in particular because it is not bounded). In the case of the ReLU non-linearity $\phi(s) = \max\{0, s\}$, there is in addition a lack of smoothness issue. An interesting direction for future research would be to adapt this algorithm and analysis in order to cover the case of ReLU non-linearities.

**Relaxing the boundedness assumption**   In Proposition 5.1, we assumed that $\Phi$ is bounded in order to obtain easy quantitative bounds on the LSI constant, but this assumption is not necessary. For instance, the Lyapunov condition in (Cattiaux et al., 2010, Cor. 2.1) implies that if there exists $C_1, C_2, C_3 > 0$ such that for all $\mu \in \mathcal{P}_2(\mathbb{R}^d)$, $V[\mu] \in \mathcal{C}^2(\mathbb{R}^d)$ and

$$\forall \|x\|_2 \geq C_1, \quad x^\top \nabla V[\mu](x) \geq C_2 \|x\|_2^2 \quad \text{and} \quad \forall x \in \mathbb{R}^d, \quad \nabla^2 V[\mu](x) \succeq -C_3 \text{Id} \tag{23}$$

then a uniform LSI holds (i.e. Assumption 3 holds).

As an illustration, consider a positively 2-homogeneous and $\mathcal{C}^2$ (uniformly in $z$) function $x \mapsto \Phi(x, z)$ (this does not cover functions of the form Eq. (22) except in the simple case $\phi = \text{id}$; a valid example is the square non-linearity $\Phi(x, z) = (x^\top z)^2$). Given the expression of $V$ in Proposition 5.1 and invoking Euler's identity $x^\top \nabla \Phi(x, z) = 2\Phi(x, z)$, we have

$$x^\top \nabla V[\mu](x) = 2\mathbf{E}_{(z,y)\sim\mathcal{D}} \, \ell'\Big(y, \int \Phi(z, x')\mathrm{d}\mu(x')\Big)\Phi(z, x) + \lambda \|x\|^2.$$

So, assuming $\ell'$ is absolutely bounded by some $M > 0$, the first part of Eq. (23) holds for $\lambda$ large enough, namely $\lambda > 2M\mathbf{E}|\Phi(z, u)|$ for all $u \in \mathbb{S}_{d-1}$. Moreover, the Hessian lower-bound is also satisfied since $\nabla^2 \Phi$ is 0-homogeneous. Thus, in this simple example, Assumption 3 holds although $\Phi$ is unbounded with a quadratic growth (but this is at the price of requiring a strong weight decay regularization).

### 5.2 Numerical illustration: kernel Maximum Mean Discrepancy

We conclude this paper with numerical experiments exploring the behavior of NPGD[1] defined in (3). Let us stress that our theoretical guarantees only apply to the mean-field Langevin dynamics – recovered in the many-particle and continuous time limit – so there remains a gap between the theory and the NPGD algorithm.

We consider the torus $\mathcal{X} \coloneqq (\mathbb{R}/(2\pi\mathbb{Z}))^d$ and the convex function defined on $\mathcal{P}(\mathcal{X})$ by

$$G(\mu) \coloneqq \frac{1}{2} \int k(x,y) \mathrm{d}\mu(x)\mathrm{d}\mu(y) - \int k(x,y)\mathrm{d}\mu(x)\mathrm{d}\nu(y) + \frac{1}{2}\int k(x,y)\mathrm{d}\nu(x)\mathrm{d}\nu(y) \tag{24}$$

where $k \in \mathcal{C}^2(\mathcal{X} \times \mathcal{X})$ is a smooth positive semi-definite kernel and $\nu \in \mathcal{P}(\mathcal{X})$ a fixed probability measure. This function $G$ can be interpreted as the square kernel Maximum Mean Discrepancy (kMMD) (Gretton et al., 2008) between $\mu$ and $\nu$. This choice of function is convenient for numerical experiments because its minimum value is known and is 0, attained in particular for $\mu = \nu$. Although our theory was developed for $\mathcal{X} = \mathbb{R}^d$, it is straightforward to adapt it to the torus, and our main convergence results apply, as shown below.

**Proposition 5.2.** *The first variation of $G$ is given, for $\mu \in \mathcal{P}(\mathcal{X})$ and $x \in \mathcal{X}$ by*

$$V[\mu](x) = \int_{\mathcal{X}} k(x,y)\mathrm{d}(\mu - \nu)(y).$$

*Moreover, Assumptions 1,2 and 3 are satisfied with $\rho_\tau \geq ((1+2\pi)d)^{-1}e^{(\inf k - \sup k)/\tau}$.*

*Proof.* The properties of $G$ and $V$ are obtained by standard arguments. For the log-Sobolev inequality, we note that the normalized volume measure on $\mathcal{X}$ satifies LSI with a constant larger than $\frac{\pi^2}{(1+2\pi)\mathrm{diam}(\mathcal{X})^2}$ (Ledoux, 1999, Thm. 7.3) and here $\mathrm{diam}(\mathcal{X}) = \pi\sqrt{d}$. The lower bound on $\rho_\tau$ follows by the Holley-Stroock criterion. $\qquad\square$

In our experiments, we consider $d = 2$ and the translation invariant kernel $k(x,y) = \prod_{i=1}^d \left(1 + 2\sum_{k=1}^n (1+k)^{-1}\cos(k(x_i - y_i))\right)$ with $n = 5$ frequency components. Because of the frequency cut-off, this kernel is not strictly positive definite, so $G$ admits minimizers other than $\nu$ (Sriperumbudur et al., 2010; Simon-Gabriel et al., 2020) although in practice we observed that NPGD was in general attracted towards $\nu$. We take $\nu$ as a random empirical distribution of $m^* = 10$ samples from the uniform distribution on $\mathcal{X}$. We run NPGD with $m = 50$ particles, a step-size $\eta = 0.08$ and $\mu_0$ being the uniform distribution on $\mathcal{X}$.

Figure 1a shows an example of a large-time particle configuration, with the atoms of $\nu$ is red and the atoms of $\hat{\mu}_t$ in black (with $t$ large), with a noise temperature $\tau = 0.1$. Here the measure $\hat{\mu}_t$ is a noisy version of $\nu^*$.

Figure 1b shows the evolution of the objective $F_\tau = G + \tau H$ (up to a constant, adjusted for ease of comparison) along the iterations, where the entropy $H$ is estimated using the 1-nearest-neighbor estimator (Kozachenko and Leonenko, 1987; Singh et al., 2003). We observe the exponential decay of $F_\tau$ towards a plateau which we expect to be the global minimum of $F_\tau$, up to discretization errors. For small values of $\tau$, it is not excluded that the plateau corresponds instead to a suboptimal metastable state.

Finally, Figure 1c shows the advantage of NPGD with simulated annealing vs. PGD to minimize the unregularized function $G$. We used a noise temperature that decays polynomially as $\tau_t = 20(t+1)^{-1}$ where $t$ is the iteration count, which is a faster decay than what the theory suggests. At iteration 800, we stopped the noise in order to observe the "quality" of the configuration of particles. We see that the NPGD with simulated annealing consistently outperforms PGD, which gets stuck in poorer local minima.

## 6 Conclusion

We have proved the convergence of the Mean-Field Langevin dynamics to the global minimizer at an exponential rate, under natural assumptions that include all settings where (non-quantitative) convergence

---

[1]Link to Julia code to reproduce the experiments: `https://github.com/lchizat/2022-mean-field-langevin-rate`.

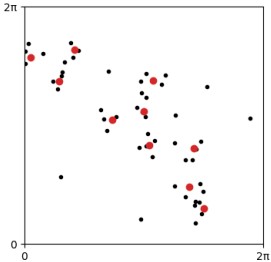
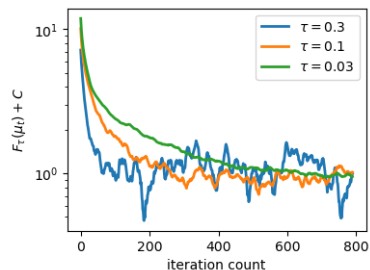
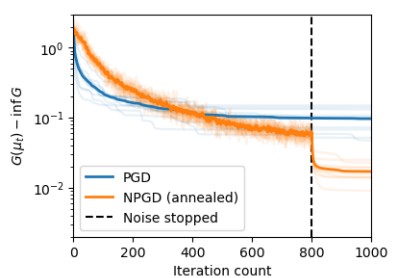

(a) Example of a large-time config-uration of NPGD for $\tau = 0.1$. (red) atoms of $\nu$ (black) atoms of $\hat{\mu}_t$.

(b) Evolution of $F_\tau(\mu_t)$ (aver-aged over 10 random experiments). Curves' height adjusted to end at 1.

(c) Evolution of $G(\mu_t)$ for NPGD with simulated annealing vs. PGD (averaged over 10 experiments).

was previously shown. We have also proved the convergence of the annealed dynamics for a suitable noise decay.

From a higher perspective, our analysis—in particular the simple "entropy sandwich" Lemma 3.4—suggests that often, the guarantees about Langevin dynamics obtained via log-Sobolev inequalities can be generalized to *mean-field* Langevin dynamics. In this paper, we focused on exponential convergence and on simulated annealing, but other aspects could be considered, such as a direct analysis of the discrete dynamics, which could lead to computational bounds, as done in e.g. (Vempala and Wibisono, 2019; Ma et al., 2019) for the Langevin algorithm.

Another interesting direction for future work is to develop and study more applications of Mean-Field Langevin dynamics, since many problems can be cast as optimization problems of the form Eq. (1). This includes sparse deconvolution problems, mixture models fitting (Boyd et al., 2017) or problems involving optimal transport (Peyré and Cuturi, 2019, Chap. 9).

### Acknowledgments

I would like to thank Loucas Pillaud-Vivien for orienting me through the literature of simulated annealing and Mo Zhou for noticing a gap in a previous version of the proof of Theorem 4.1. I would also like to thank the anonymous reviewers for their insightful comments and for suggesting the discussion on LSI in Section 5.1.

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

## A    Additional proofs

Let us start with a relation between $G$ and its first-variation $V$ that is more convenient for proofs.

**Lemma A.1** (Integral formula). *Under Assumption* (1), *for $\mu_0, \mu_1 \in \mathcal{P}_2(\mathbb{R}^d)$, one has*

$$G(\mu_1) - G(\mu_0) = \int_0^1 \int_{\mathbb{R}^d} V[\mu_t]\mathrm{d}(\mu_1 - \mu_0)\mathrm{d}t$$

*where $\mu_t = (1-t)\mu_0 + t\mu_1$ for $t \in [0,1]$.*

*Proof.* Let $h(t) = G(\mu_t)$. By definition of the first-variation, $h$ is right (resp. left) continuous at $t = 0$ (resp. $t = 1$). We just need to prove that $h$ is differentiable on $]0,1[$ with $h'(t) = \int V[\mu_t]\mathrm{d}(\mu_1 - \mu_0)$. Then, because this expression is continuous in $t$ under Assumption 1, the fundamental theorem of calculus would imply $h(1) - h(0) = \int_0^1 h'(t)\mathrm{d}t$, which is our claim. For $t, \epsilon \in ]0,1[$ one has $(1-\epsilon)\mu_t + \epsilon\mu_0 = \mu_{t-t\epsilon}$ and thus

$$-th'_-(t) = \lim_{\epsilon \to 0_+} \frac{h(t-t\epsilon) - h(t)}{\epsilon} = \lim_{\epsilon \to 0+} \frac{G((1-\epsilon)\mu_t + \epsilon\mu_0) - G(\mu_t)}{\epsilon}$$

$$= \int V[\mu_t]\mathrm{d}(\mu_0 - \mu_t) = -t \int V[\mu_t]\mathrm{d}(\mu_1 - \mu_0)$$

where $h'_-(t)$ stands for the left-derivative of $h$ at $t$. This shows that $h'_-(t) = \int V[\mu_t]\mathrm{d}(\mu_1 - \mu_0)$ for $t \in ]0,1[$. A similar computation using $(1-\epsilon)\mu_t + \epsilon\mu_1 = \mu_{t+(1-t)\epsilon}$ shows that the right derivative $h'_+(t)$ has the same value, and thus $h'(t) = \int V[\mu_t]\mathrm{d}(\mu_1 - \mu_0)$ for $t \in [0,1]$ which concludes the proof of the formula. □

In the following lemma, we verify that $G$ is well-behaved (in fact smooth) as function in the Wasserstein space $\mathcal{P}_2(\mathbb{R}^d)$, using the vocabulary and results from Ambrosio and Savaré (2007).

**Lemma A.2.** *Let $\mu \in \mathcal{P}_2(\mathbb{R}^d)$, let $v \in L^2(\mu)$ and let $\mu_t = (\mathrm{id} + tv)_\#\mu$. Then*

$$\frac{d}{dt}G(\mu_t) = \int_{\mathbb{R}^d} \nabla V[\mu_t](x + tv(x))^\top v(x)\mathrm{d}\mu(x).$$

*Moreover, $G$ is $(-2L)$-semiconvex along any interpolating curve in $\mathcal{P}_2^a(\mathbb{R}^d)$ and the $W_2$-derivative of $G$ at $\mu$ is $V[\mu]$.*

Since the same holds true for $-G$, we could say that $G$ is $2L$-smooth in the Wasserstein geometry, in the sense that it is both $2L$-semiconvex and $2L$-semiconvex.

*Proof.* For $\epsilon > 0$ and $s \in [0,1]$, let $\mu_s = (1-s)\mu + s\mu_{t+\epsilon}$. It holds by Lemma A.1,

$$\frac{1}{\epsilon}(G(\mu_{t+\epsilon}) - G(\mu_t)) = \frac{1}{\epsilon}\int_0^1 \int_{\mathbb{R}^d} V[\mu_s]\mathrm{d}(\mu_{t+\epsilon} - \mu_t)$$

$$= \frac{1}{\epsilon}\int_0^1 \int_{\mathbb{R}^d}(V[\mu_s](x + (t+\epsilon)v(x)) - V[\mu_s](x + tv(x)))\mathrm{d}\mu(x)$$

$$= \int_0^1 \int_{\mathbb{R}^d} \nabla V[\mu_s](x + tv(x))^\top v(x)\mathrm{d}\mu(x) + O(L\epsilon\|v\|_{L^2(\mu)})$$

$$= \int_{\mathbb{R}^d} \nabla V[\mu](x + tv(x))^\top v(x)\mathrm{d}\mu(x) + O(L\epsilon\|v\|_{L^2(\mu)})$$

where we used successively the Lipschitz continuity of $x \mapsto \nabla V[\mu](x)$ and of $\mu \mapsto \nabla V[\mu](x)$ in the last two lines. The first claim follows by taking the limit $\epsilon \to 0$. This also shows that $V[\mu]$ is the unique (strong) $W_2$-differential of $W_2$ at $\mu$, in the sense of (Ambrosio and Savaré, 2007, Def. 4.1).

For the semi-convexity claim, let $h(t) := G(\mu_t)$. For $s, t \in [0, 1]$, it holds by Cauchy-Schwarz

$$|h'(t) - h'(s)|^2 \leq \|v\|_{L^2(\mu)}^2 \int_{\mathbb{R}^d} \|\nabla V[\mu_t](x + tv(x)) - \nabla V[\mu_s](x + sv(x))\|^2 \mathrm{d}\mu(x)$$

$$\leq \|v\|_{L^2(\mu)}^2 L^2 \int_{\mathbb{R}^d} \left(W_2(\mu_s, \mu_t) + |t - s|\|v(x)\|\right)^2 \mathrm{d}\mu(x)$$

$$\leq \|v\|_{L^2(\mu)}^2 L^2 (2W_2^2(\mu_s, \mu_t) + 2|t - s|^2 \|v\|_{L^2(\mu)}^2).$$

Since $W_2(\mu_s, \mu_t) \leq |t - s|\|v\|_{L^2(\mu)}$, it follows

$$|h'(t) - h'(s)| \leq 2L|t - s|\|v\|_{L^2(\mu)}^2$$

which proves that $G$ is $(-2L)$-convex in the sense of (Ambrosio and Savaré, 2007, Remark 3.2). $\qquad\square$

