# OpenReview forum: "Mean-Field Langevin Dynamics : Exponential Convergence and Annealing"
_TMLR — Accepted by TMLR_

### Review · Reviewer_QWcw · 2022-06-11

**Summary Of Contributions:**

This submission studies the convergence rate of a generalization of the the Langevin dynamics termed the *mean-field Langevin dynamics*, which can be viewed as an optimization algorithm in the space of measures that minimizes an entropy-regularized convex functional. This dynamics naturally arises from the optimization of two-layer neural network (in the mean-field regime) using the noisy gradient descent algorithm. The main contribution of this submission is a simple proof that the continuous-time dynamics converges exponentially to the optimum, under a uniform log-Sobolev inequality (LSI) on the trajectory. In addition, an annealed dynamics with decreasing entropy regularization is also studied and a slower convergence rate is obtained.

**Broader Impact Concerns:**

This submission does not require a Broader Impact Statement.

**Requested Changes:**

Overall, I feel that this is an interesting submission that is relevant to the TMLR community. I hope the authors can comment on the weaknesses mentioned above, and be a bit more explicit about the limitations of the current approach.

**Strengths And Weaknesses:**

### **Strength:**

- global convergence of two-layer neural network in the mean-field regime is an important research problem, and most existing works do not prove a quantitative convergence rate, even in the presence of regularization. This submission fills this gap by proving linear convergence for the regularized objective.

- The analysis is based on a clever observation that the optimality gap can be "sandwiched" between two relative entropy terms, one of which relates to the Gibbs measure defined by the objective "linearized" at the current point. This observation, together with the uniform LSI condition, gives a simple proof of the convergence rate.

- I expect that the studied mean-field Langevin dynamics can find applications beyond two-layer neural networks, since it covers settings where sampling via the vanilla Langevin dynamics is not possible.

### **Weakness:**

- The convergence rate is only shown for the continuous-time and infinite-particle setting, but the algorithm employed in practice requires space- and time-discretization. While the concurrent [Nitanda et al. 2022] provided a discrete-time analysis, the finite-particle guarantee is still not available.
This is especially challenging for the annealed dynamics, where the very slow convergence rate likely results in the discretization error blowing up through time.

- In the context of two-layer neural network, the LSI constant has exponential dependence on the regularization hyperparameter. This limits the applicability of the theory in providing meaningful convergence rate under weak regularization (which is important if we want to cover the near-interpolation setting or exact recovery in student-teacher settings).

- Related to the previous point, the uniform LSI condition is shown by invoking the Holley-Stroock criterion. While this derivation may be optimal in some settings, it also feels a bit underwhelming: the neural network function itself is treated as a "perturbation", and in some sense training converges only because of the regularization.

---

### Review · Reviewer_A4WX · 2022-06-13

**Summary Of Contributions:**

The paper studies a convergence of mean-field Langevin dynamics which is a time-and-many particle limit of noisy gradient descent for mean-field models.  In particular, it is shown that this dynamics minimizes the convex functional plus negative entropy on the space of measures and converges to the optimal solution with an exponential rate.
This result includes an important application: the global convergence of the optimization of two-layer neural networks in the mean-field regime.
Moreover, the paper also studies annealing dynamics in which the noise decays at a logarithmic rate, which implies the global convergence to the solution for unregularized problems.

**Broader Impact Concerns:**

Not applicable.

**Requested Changes:**

A few concerns (especially the first one) about the proof commented above should be addressed and corrected if needed.

Below are some minor comments:

- About the comment just before Sec 1.1.: ``As for the behavior of the Mean-Field Langevin dynamics (5) itself, it is shown ... but these works leave open the question of quantitative".
It is better to note that the exponential convergence in $W_1$-distance was shown in related studies. For instance, see Remark 2.12 in [Hu et al. (2019)].  However, the contribution of the paper is not negated by the previous work, because they required the strong regularization.

- Typo: $h(\tau)=F_{\tau_t}(\mu_\tau^*)$ on line 3 (page 8) is correctly $h(\tau)=F_{\tau}(\mu_\tau^*)$.

- Typo: $C_2$ (which is the last term in the (displayed) second inequality on page 8) is correctly $C_3$.

**Strengths And Weaknesses:**

Contributions:

- Recently, convergence analysis of the mean-field Langevin dynamics has become a very important research topic because of the preferable property of neural networks in the mean-field regime. In fact, many studies analyzed the convergence of the dynamics.
However, previous related works only show the convergence to the unique minimizer without an explicit convergence rate or require strong regularization to guarantee the quantitative convergence.
In contrast, the paper succeeds in providing an explicit convergence rate under any strength of regularization with a much more simple analysis.

- Moreover, the proof is quite neat and will be useful for the community because it is a natural extension of the analysis of normal Langevin dynamics which is recovered as a special case when the objective is linear functional plus entropy. This proof technique will be further investigated in the future.

- I would acknowledge the analysis of annealing dynamics is a unique and important contribution, although the result on non-annealing dynamics was also obtained by the concurrent work [Nitanda et al. 2022].


While these contributions, I think the proof of Theorem 4.1 probably needs to be fixed a bit as commented below.
If my understanding is wrong, I would appreciate it if you could correct me.


- Doesn't the constant $C_2$ (in the first inequality on page 8) depend on $t$?
    From Lemma 4.2, $-\tau_t H(\mu_t)$ is upper bounded by $\tau_t C_1 F_{\tau_t}(\mu_t) + \tau_tC_2$. Indeed, the second term can be free from $t$ because of monotonicity of $\tau_t$ and positivity of $C_2$, but that is not the case for the first term because $F_{\tau}(\mu)$ can be negative. Please elaborate on how $\tau_t C_1$ of $\tau_t C_1 F_{\tau_t}(\mu_t)$ can be independent of $t$ in the inequality.

- Minor comment: I am not sure how the term $C' t^{-1-\epsilon}$ (which is the last term in the (displayed) fourth inequality on page 8) is derived. I guess this constant relates to $t^{-1}/\log t$ in the previous inequality, but the inequality is reversed. I think it can be corrected: the term $t^{-1}/\log t$ could remain itself, which does not affect the argument of Step 1 because $t^{-1}/\log t$ is dominated by $-t^{-1}$ in Eq.(20).

- Suggestion: It would be helpful for the readers to elaborate a bit more on the derivation of the first inequality on $| G(\tilde{\mu_0}) - G(\tilde{\mu_\sigma})|$ (appearing on the first displayed inequality on page 9), though I understand ``Gaussian kernel is centered" is used to show $\int \nabla V(\tilde{\mu}_\sigma)(x)^\top (y-x)d\gamma(x,y)=0$ through $\int \nabla V(\tilde{\mu}_\sigma)(x)^\top z g_\sigma(z)dz=0$.

---

### Review · Reviewer_piEG · 2022-06-19

**Summary Of Contributions:**

This paper studies the convergence rate of mean-field Langevin dynamics (MFLD). The main contributions lie in three folds: (i). the authors prove that under a certain family of LSI holds, the MFLD converges at an exponential rate; (ii). the authors study the convergence of annealed dynamics and prove when the noise decay at an logarithmic rate, the dynamics also converges for the unregularized objective; (iii). for the application of (i), the author study the mean-field dynamics of two-layer neural networks and obtain the exponential convergence rate to the global optima.


**Broader Impact Concerns:**

N.A.

**Requested Changes:**

1. discuss more related works including what mentioned in [Strengths And Weaknesses].

2. [not critical] may run some toy experiments to make the theory more convincing.

3. [not critical] discuss some limitation of assumptions, for example, in proposition 5.1 of  |E[y|z]| \leq K'.

**Strengths And Weaknesses:**

Strength:
The paper is clearly writing and well-organised.  The idea of "entropy sandwich" is intuitively interesting and theoretic promising, although it is a direct application of convexity of functionals. The application of obtaining the convergence rate of two-layer neural networks training in the mean-field regime is also an important result.

Weakness:
I have some minor concerns on the discussion of the related works. The exponential convergence of non-linear Langevin dynamics seems already obtained in previous works. For example, in [Yang, Zhuoran, et al. 2020], the authors proved a similar exponential convergence result of mean-field particle optimization in Theorem 4.8 under a similar "uniform" Polyak-Lojasiewicz condition in Assumption 4.6 and Equation 4.14.  I think the authors should provide some discussions on the similarity and difference from previous works. In particular, how Theorem 3 and Assumption 3 differs from [Yang, Zhuoran, et al. 2020].

[not critical] In proposition 5.1, when \ell is the square loss, there should be an additional assumption that |E[y|z]| \leq K'. I'm a little bit confused with this assumption: does it mean we should put some restrictions on the data distribution D? If it is, this assumption seems to be somewhat restrictive.


[not critical] There seems no experimental validations. It is good even the author could run some toy examples to validate the theory.

Reference:

[1]. Yang, Zhuoran, et al. "Variational Transport: A Convergent Particle-Based Algorithm for Distributional Optimization." arXiv preprint arXiv:2012.11554 (2020).

---

> ### Author Response · Authors · 2022-06-27
> **We have added numerical experiments in the revised version**
>
> We would like to thank the reviewer for his time and feedback.
>
> - We have added toy numerical experiments in Section 5.2, in the context where $G$ is a squared Maximum-Mean-Discrepancy. We also emphasized on the fact that our theory does not directly cover the discretized dynamics with a finite number of particles; hence these numerical experiments may exhibit phenomena that are not covered by the theory.
>
> - Concerning reference [1]: there is a fundamental difference between [1] and our contribution: in [1] the authors *assume* a PL inequality and deduce convergence while in our paper we *prove* that such an inequality holds in our particular context. The fact that a PL inequality implies exponential convergence is known in very general contexts (see in particular [2] for the case of Wasserstein gradient flows). One major challenge is to find natural assumptions under which such inequalities can be shown to hold, which is one of our contributions.
>
> - Yes, the assumption $|E[y|z]| \leq K'$ puts restriction on the data distribution: the Bayes predictor should be bounded. We believe that this assumption is less unusual than the boundedness of $\phi$, this is why we do not insist much on it.
>
>
> [2] A family of functional inequalities: Łojasiewicz inequalities and displacement convex functions, A. Blanchet, J. Bolte - Journal of Functional Analysis, 2018

---

> > ### Comment · Reviewer_piEG · 2022-07-03
> > **Thanks for detailed response**
> >
> > I'm satisfied with authors' response and lean to accept the paper.
> >
> > However, I still have some concerns on *finding natural assumptions* under which LSI holds for *two-layer neural networks*.
> >
> > My questions is whether the *boundedness* assumption is a *necessary* condition to establish LSI for *two-layer neural networks*:
> >
> > This paper establishes uniform LSI for two-layer neural networks by imposing boundedness assumptions on activations and then by Holley-Stroock perturbation lemma where the neural network function is a bounded perturbation of the L2 regularizer that trivially satisfies LSI.  As pointed out by another reviewer, in some sense training converges only because of the regularization. Furthermore, the boundedness assumption does not hold for the standard two-layer neural networks where $\Phi(z, x) = x_2 h(x_1, z)$ for $x_2\in \mathbb{R}$.
> >
> > There seems many other ways to deduce LSI beyond what the paper adopts "strongly convex L2 regularizer + bounded neural network perturbation". A general principle is to control the curvature lower bound of the nonlinear drift. In this sense, the boundedness assumption for two-layer neural networks seem not satisfactory enough and not a necessary condition to establish the LSI. Perhaps some local smoothness and growth condition of $\Phi(z, x) = x_2 h(x_1, z)$ on $||x||_2$ is sufficient to establish the LSI.

---

> > > ### Author Response · Authors · 2022-07-07
> > > **NN assumption can be weakened but at the price of a stronger regularization**
> > >
> > > For the application to neural networks, this paper only covers *noisy* gradient descent and it is thus clear that it is the noise (i.e. the entropic regularization) that makes the dynamics converge globally (with our proof technique). This remains true even if we relax the assumptions on the activation function.
> > >
> > > Now the objective of the paper is not to study in details when the LSI assumption holds for neural networks (which are mostly mentioned as an illustration of the result, along with kernel MMD), which is why we presented only one simple example where LSI holds, with simple quantitative bounds. The LSI can be obtained under weaker assumptions in this context, but there is some obstructions to cover "natural" two-layer neural networks with weight decay regularization, which is why we did not push in this direction.
> > >
> > > One simple criterion that guarantees that a smooth potential $V:\mathbb{R}^d\to \mathbb{R}$ satisfies LSI is $\lim_{R\to \infty}\inf_{\Vert x\Vert\geq R} \nabla^2 V(x)>0$ [1]. This is a rather sharp condition, and it inspecting the proof in [1] it can be seen that it can also lead to a uniform LSI if the infimum is also taken over a family of potentials. So all that is needed is strong convexity of $V$ at infinity.
> > > If $V$ contains a $\lambda$-strongly convex potential term (as is our case), we are thus allowed to add a perturbation as long as its Hessian is strictly bounded by $\lambda$ at infinity. This covers for instance two-layer neural nets with a nonlinearity $\sigma$ such that $\sigma''$ is bounded, and with *fixed output weights, provided $\lambda$ is large enough* (plus some integrability conditions on the input data).  One could alternatively consider a convex regularizer with a Hessian that grows fast enough to $\infty$ as $\Vert x\Vert\to x$, such as $\lambda \Vert x\Vert^{p}$ for $p>3$, which gives more "room" to the non-convex perturbation and with this choice, vanilla two-layer neural networks with $\sigma''$ bounded are allowed.
> > >
> > > Overall, weakening the assumptions on the nonlinearity requires to strengthen the regularization. We will include this discussion (with more details) in the revision of the paper.
> > >
> > > [1] Feng-Yu Wang, Logarithmic Sobolev Inequalities: conditions and counterexamples.

---

### Decision · Action_Editors · 2022-07-20

**Recommendation:** Accept as is

**Comment:**

All three reviewers and the AE think that this a good paper with solid contributions. The author also have addressed a few minor concerns which improved the quality of the paper even further.

The author should address any remaining minor issues (e.g. the promised discussion on weakened assmp w/ stronger regularization) before camera ready.

AE

---

> ### Author Response · Authors · 2022-08-08
> **thanks**
>
> Thanks you for organizing this review process of great quality !
> I just submitted the (de-anonymized) camera ready version of the paper (which includes the promised discussion on LSI and a few more references to the mathematical physics literature).
> I intend to update the arxiv version of this paper to this camera ready version. I do not see any rule about this on the TMLR website, please tell me if there is any issue with that!

---

> > ### Comment · Action_Editors · 2022-08-10
> > **Congratulations!**
> >
> > I am confirming that it is OK to post your camera ready version on arxiv.
> > -AE